# Preparation of Magnetic–Luminescent Bifunctional Rapeseed Pod-Like Drug Delivery System for Sequential Release of Dual Drugs [note 1]

**DOI:** 10.3390/pharmaceutics13081116

**Published:** 2021-07-22

**Authors:** Junwei Xu, Yunxue Jia, Meili Liu, Xuenan Gu, Ping Li, Yubo Fan

**Affiliations:** 1Key Laboratory for Biomechanics and Mechanobiology of Ministry of Education, Beijing Advanced Innovation Center for Biomedical Engineering, School of Biological Science and Medical Engineering, Beihang University, Beijing 100191, China; junweixu@buaa.edu.cn (J.X.); sharon.jia@briconmedical.cn (Y.J.); liuml@buaa.edu.cn (M.L.); xngu@buaa.edu.cn (X.G.); 2School of Medical Science and Engineering, Beihang University, Beijing 100191, China

**Keywords:** rapeseed pod-like drug delivery system, electrospray, magnetic–luminescent bifunctional drug delivery carrier, dual drugs, sequential release

## Abstract

Drug delivery systems (DDSs) limited to a single function or single-drug loading are struggling to meet the requirements of clinical medical applications. It is of great significance to fabricate DDSs with multiple functions such as magnetic targeting or fluorescent labeling, as well as with multiple-drug loading for enhancing drug efficacy and accelerating actions. In this study, inspired by the dual-chamber structure of rapeseed pods, biomimetic magnetic–luminescent bifunctional drug delivery carriers (DDCs) of 1.9 ± 0.3 μm diameter and 19.6 ± 4.4 μm length for dual drug release were fabricated via double-needle electrospraying. Morphological images showed that the rapeseed pod-like DDCs had a rod-like morphology and Janus dual-chamber structure. Magnetic nanoparticles and luminescent materials were elaborately designed to be dispersed in two different chambers to endow the DDCs with excellent magnetic and luminescent properties. Synchronously, the Janus structure of DDCs promoted the luminescent intensity by at least threefold compared to single-chamber DDCs. The results of the hemolysis experiment and cytotoxicity assay suggested the great blood and cell compatibilities of DDCs. Further inspired by the core–shell structure of rapeseeds containing oil wrapped in rapeseed pods, DDCs were fabricated to carry benzimidazole molecules and doxorubicin@chitosan nanoparticles in different chambers, realizing the sequential release of benzimidazole within 12 h and of doxorubicin from day 3 to day 18. These rapeseed pod-like DDSs with excellent magnetic and luminescent properties and sequential release of dual drugs have potential for biomedical applications such as targeted drug delivery, bioimaging, and sustained treatment of diseases.

## 1. Introduction

Drug delivery systems (DDSs) were proposed to improve the efficiency of drug utilization and reduce toxicity [1,2]. With the development of DDSs, improving the targeting efficiency of drug delivery carriers (DDCs) has attracted widespread attention. DDSs were expected to be endowed with multiple functions in terms of targeting control [3,4]. DDSs constructed with magnetic and luminescent materials achieve both a magnetic response and fluorescent labeling, with applications in magnetic targeting, bioimaging, and drug delivery [5,6]. Guo et al. [7] encapsulated fluorescent dye Cy5.5, methotrexate, and oleic acid-modified magnetic nanoparticles (MNPs) into liposomes, inducing a dual-imaging effect to verify liposome accumulation in the tumor region. However, the presence of a magnetic material tends to quench the luminescent material and reduce the luminescence intensity [8,9]. Specific strategies were carried out to avoid the quenching effect, such as constructing a core–shell structure or double-chamber structure to separate magnetic materials from luminescent materials [10,11]. Yang et al. [10] encapsulated Fe_3_O_4_ MNPs with nonporous silica and a further layer of ordered mesoporous silica through a sol–gel process and further functionalized the outer silica shell by deposition of YVO_4_:Eu^3+^ phosphors, achieving a magnetic coating separate from the inside luminescent material. Wang et al. [12] fabricated poly(vinyl pyrrolidinone) Janus nanofibers with one side embedding Fe_3_O_4_ MNPs and the other side containing [Eu(BA)_3_phen+Tb(BA)_3_phen] (BA = benzoic acid, phen = 1,10-phenanthroline) by parallel electrospinning, resulting in high luminescence intensity almost unaffected by Fe_3_O_4_ MNPs.

In addition to using magnetic or fluorescent materials to enhance targeting efficiency, the regulation of morphology, size, and surface composition of carriers also affects the therapeutic effect of DDSs [13]. For example, using liposomes as DDCs can help the DDSs reach the lesions more efficiently [14]. The strategy of preparing rod-shaped or urchin-like particles was also proposed to promote cell phagocytosis efficiency by increasing the contact of DDSs with cells [15,16]. According to an in-depth study of non-spherical particles, rod-shaped DDSs were found to have a better effect than common spherical DDSs in the treatment of cancer and other diseases because of the large specific surface area, high drug-loading capacity, and long circulation time [17,18,19,20]. Xu et al. [21] indicated that mesoporous silica nanorods loaded with indomethacin showed an approximately 1.3-fold longer blood circulation in the body and 2.2-fold higher bioavailability than nanospheres, and the nanorods could more easily overcome rapid clearance by the reticuloendothelial system in comparison to nanospheres. Zhang et al. [22] prepared fiber rods of 2–10 μm in length for cancer treatment by breaking electrospun fibers with ultrasonic fragmentation using NaCl as a porogen. Compared to microspheres, fiber rods exhibited an up to fourfold higher accumulation in tumors and an up to threefold longer terminal half-life of plasma drug levels after intravenous injection.

Currently, single-drug loading is struggling to meet treatment needs due to the complexity of the human body and the internal environment [23,24,25]. It is of great significance to fabricate DDSs loaded with multiple drugs, especially the sequential release of different drugs, to enhance the treatment efficacy [26,27,28,29]. It has been reported that the sequential release of doxorubicin (DOX) and bromodomain-containing protein 4 inhibitor (BRD4i) JQ1 remarkably inhibited growth and suppressed lung metastasis of 4T1 breast tumor [30]. The strategy of a rapid release of vancomycin for the initial 2 days, followed by sustained release of recombinant human bone morphogenetic protein-2, was used to inhibit bacterial invasion at the initial stage and later promote the behavior of related osteogenesis cells for favorable osteointegration after dental implantation [31]. Apparently, DDSs with a more complex structure are required to induce the sequential release of multiple drugs, such as core–shell micro/nanoparticles, and fibers [32,33,34,35]. Nie et al. [36] prepared poly(l-lactide)/poly(lactide-*co*-glycolide) (PLGA) core–shell microspheres by coaxial electrohydrodynamic atomization, in which suramin and paclitaxel were loaded in the core and shell layers, respectively. Li et al. [35] developed a methoxy poly(ethylene glycol)-*block*-PLGA/dextran core–sheath nanofiber patch which contained hydrophobic 10-hydroxycamptothecin and hydrophilic tea polyphenols in the shell and core of the nanofiber, respectively, realizing the sequential release of the two drugs. Similarly, entrapping drug-loaded nanoparticles (NPs) into fibers or hydrogels is also an important strategy.

In nature, rapeseed pods exhibit a long dual-chamber structure with rapeseeds containing oil wrapped in the two chambers of rapeseed pods. Inspired by the special structure of rapeseed pods (Figure 1A), magnetic–luminescent bifunctional rapeseed pod-like DDCs ([NaYF_4_:Eu^3+^/chitosan (CS)@PLGA]//[Fe_3_O_4_@PLGA]) were produced by double-needle electrospraying. One chamber contained luminescent NaYF_4_:Eu^3+^ NPs and CS NPs, whereas magnetic Fe_3_O_4_ MNPs were loaded in another chamber of the DDCs. The dual-chamber structure was expected to reduce the interaction of magnetic material with luminescent material and, thus, enhance luminescent intensity. In the present study, the morphology, structural, magnetic, and luminescence properties, as well as the biocompatibility, of the rapeseed pod-like DDCs were studied in detail. To explore the drug release performance, dual-drug-loaded rapeseed pod-like DDSs ([NaYF_4_: Eu^3+^/(DOX@CS)@PLGA]//[benzimidazole (Bim)/Fe_3_O_4_@PLGA]) were fabricated by loading the model drug DOX embedded in CS NPs, as well as another drug Bim in the magnetic chamber. Bim and DOX@CS NPs scattered in the double chambers could realize the combined application, and the release of the dual drugs was evaluated by drug release kinetics and cell experiments in vitro. Furthermore, the sustained release of drugs in rapeseed pod-like DDSs showed great utility in continuously inhibiting tumor cell growth.

## 2. Materials and Methods

### 2.1. Materials

PLGA (*Mw* = 20 kDa, LA:GA = 50:50 *mol/mol*) was purchased from Jinan Daigang Biotechnology Co., Ltd. (Jinan, China). Tetrahydrofuran (THF, ≥99.0%) was bought from Macklin Biochemical Technology Co., Ltd. (Shanghai, China). FeCl_3_·6H_2_O (≥99.0%), ethylene glycol, and anhydrous sodium acetate (CH_3_COONa, NaAc) were provided by Xilong Science Co., Ltd. (Shantou, China). Y(NO_3_)_3_·6H_2_O (99.0%), polyethyleneimine (PEI), and NaBF_4_ were supplied by Shandong Xiya Chemical Industry Co., Ltd. (Linyi, China). Eu(NO_3_)_3_·6H_2_O (99.99%) and DOX (≥98.0%) were bought from Aladdin Industrial Corporation (Shanghai, China). NaCl, chitosan (CS, degree of deacetylation: 80–95%, viscosity: 0.05–0.8 Pa·s), and benzimidazole (Bim) were provided by Sinopharm Chemical Reagent Co., Ltd. (Shanghai, China). Diethylene glycol (DEG) was acquired from the Tianjin Guangfu Fine Chemicals Institute (Tianjin, China). Acetic acid (CH_3_COOH, AcOH) was received from Tongguang Fine Chemicals Company (Beijing, China). Absolute ethyl alcohol (≥99.7%) was purchased from Beijing Oriental Technology Development Co., Ltd. (Beijing, China). Deionized water was used in all experiments. All reagents involved in this work were of analytical grade and directly used as received without further purification.

### 2.2. Preparation of Rapeseed Pod-Like DDCs

Inspired by rapeseed pods, DDCs ([NaYF_4_:Eu^3+^/CS@PLGA]//[Fe_3_O_4_@PLGA]) with double chambers were fabricated via double-needle electrospraying.

#### 2.2.1. Synthesis of Fe_3_O_4_ MNPs

The preparation of Fe_3_O_4_ MNPs was carried out according to the method of Rao et al. [37] Specifically, FeCl_3_·6H_2_O (1.35 g) was dissolved in ethylene glycol (30 mL) to form a clear solution, followed by the addition of NaAc (3.6 g). The mixture was stirred for 0.5 h and then sealed in a Teflon-lined stainless-steel autoclave. The autoclave was heated to 200 °C and maintained for 7 h, cooling to room temperature. The precipitated black products were collected from the solution with an external magnet and washed with ethanol and distilled water. Lastly, the black products were dried in an oven under vacuum at 50 °C for 24 h.

#### 2.2.2. Synthesis of NaYF_4_:Eu^3+^ NPs

NaYF_4_:Eu^3+^ NPs were synthesized using a hydrothermal method [38,39]. Briefly, 0.69 g of Y(NO_3_)_3_·6H_2_O, 0.09 g of Eu(NO_3_)_3_·6H_2_O, and 0.17 g of NaCl were mixed in 10 mL of deionized water under vigorous stirring, followed by adding 30 mL of DEG and 10 mL of PEI (10 wt.%) solution in the mixture. Subsequently, a solution of NaBF_4_ (0.2 g) dissolved in deionized water (5 mL) was added to the mixture, which was stirred for 1 h and then transferred into a Teflon autoclave at 180 °C for 6 h. The products were collected and washed four times with deionized water and ethanol in turn, before drying at 60 °C for 24 h.

#### 2.2.3. Preparation of CS NPs

CS NPs were prepared via electrospraying by using a 3% *w*/*v* precursor solution of CS in a blend of AcOH and H_2_O (AcOH:H_2_O = 7:3 *v*/*v*). As shown in Figure 1Ai, the precursor solution was placed in a 1 mL syringe with a stainless-steel needle. A sheet of aluminum foil was used as the collector, which was placed at 15 cm away from the needle. The flow rate of the precursor solution was controlled by a syringe pump (78-9100C, Cole Parmer, Vernon Hills, IL, USA) at 0.2 mL/h. A positive high voltage (power, DW-N303-1ACFO, Dongwen High Voltage Power Co., Ltd., Tianjin, China) of 20 kV was applied between the needle and the collector to generate stable continuous CS NPs.

#### 2.2.4. Preparation of Rapeseed Pod-Like DDCs

As shown in Figure 1Aii, PLGA, CS NPs, and NaYF_4_:Eu^3+^ NPs were dispersed in THF as precursor solution A; PLGA and Fe_3_O_4_ MNPs were dispersed in THF as precursor solution B. During the electrospraying process to obtain DDCs ([NaYF_4_:Eu^3+^/CS@PLGA]//[Fe_3_O_4_@PLGA]) (denoted as D2), solutions A and B were loaded in two syringes, connected with a double-needle system. A collector was placed 15 cm away from the needles, a positive high voltage of about 7 kV was set, and the flow rate of the precursor solutions was 0.6 mL/h. As a control, rapeseed pod-like DDCs ([NaYF_4_:Eu^3+^@PLGA]//[Fe_3_O_4_@PLGA]) (denoted as D1) which did not load CS NPs were also produced. Single-chamber DDCs ([NaYF_4_:Eu^3+^/Fe_3_O_4_@PLGA]) (denoted as S1) and ([NaYF_4_:Eu^3+^/CS/Fe_3_O_4_@PLGA]) (denoted as S2) were prepared via single-needle electrospraying (Appendix A) to further explore the influence of the double-chamber structure on the intensity of luminescence. The detailed dosages of materials for fabricating DDCs are shown in Table 1.

### 2.3. Preparation of Rapeseed Pod-Like DDSs

As the model drugs for treating cancer, Bim was scattered in the magnetic chamber and DOX was loaded in CS NPs and then scattered in the luminescent chamber so as to further study the sequential release of the dual drugs. The rapeseed pod-like DDSs ([NaYF_4_: Eu^3+^/(DOX@CS)@PLGA]//[Bim/Fe_3_O_4_@PLGA]) (denoted as R2) were prepared as shown in Figure 1B. DOX was solved in the precursor solution of CS/AcOH/H_2_O (the ratio of CS and DOX was 5:1 *w*/*w*) to produce DOX@CS NPs via electrospraying according to Section 2.2.3, which were subsequently used to prepare precursor solution A. Bim was added to precursor solution B, and the ratio of Bim and DOX was 2:1 *w*/*w*. DDSs ([NaYF_4_: Eu^3+^/DOX/CS@PLGA]//[Bim/Fe_3_O_4_@PLGA]) (denoted as R1) were also produced as a control, in which DOX was not loaded in CS NPs, so that Bim and DOX could be released simultaneously. The detailed dosages of materials for preparing R1 and R2 DDSs are shown in Table 2.

### 2.4. Characterizations

Morphologies of Fe_3_O_4_ MNPs and NaYF_4_:Eu^3+^ NPs were observed by transmission electron microscopy (JEM 1200EX, TEM, NEC Electronics Corporation, Tokyo, Japan). Morphologies of CS NPs and DDCs were observed by scanning electron microscopy (Quanta™ 250 FEG SEM, FEI, Hillsboro, OR, USA). The particle sizes were measured using the software Image J 1.48v on TEM or SEM images. Zeta potentials of Fe_3_O_4_ MNPs or NaYF_4_:Eu^3+^ NPs dispersed in water and CS NPs dispersed in ethanol were detected using a laser particle analyzer (Zetasizer Nano ZS, Malvern, UK). The compositions of samples were investigated by FTIR spectrometry (FTIR-1650, Tianjin Gang Dong Technology Development Co., Ltd., Tianjin, China). Contact angles were recorded using a contact measurement (JC2000FM, Powereach, Shanghai, China). X-ray diffraction (XRD) patterns were measured with an X-ray diffractometer (XRD, D/MAX-2500, Rigaku, Japan) at 40 kV and 200 mA. A metallographic microscope (LV100ND, Nikon, Tokyo, Japan) was used to observe the Janus structure of samples. The magnetic performances of samples were measured by a vibrating sample magnetometer (VSM, Laker Shore 7307, Forest Lake, MN, USA), and the luminescent property was detected with a fluorescence spectrophotometer (FS5, Edinburgh, UK).

### 2.5. Hemocompatibility Evaluation

A hemolysis assay was performed to evaluate hemocompatibility with anticoagulated newborn cow blood, which was purchased from Jiaozuo Lufeifan Biotechnology Co., Ltd., Jiaozuo, China. First, 1 mL of phosphate-buffered saline (PBS) solution (Beijing Solarbio Science & Technology Co., LTD, Beijing, China) was added to 0.8 mL of whole blood to form diluted whole blood. D2 DDCs were dispersed in PBS solutions with the concentrations of 3.125, 6.25, 12.5, 25, 50, 100, 200, 400, and 800 μg/mL, respectively. After that, 20 μL of diluted whole blood was added to 1 mL of D2/PBS solution at each concentration. The positive and negative controls were distilled water containing diluted whole blood and PBS, respectively. All samples were incubated at 37 °C for 30 min and then centrifuged at 2500 rpm to obtain supernatants. The absorbance of the supernatants was measured at 545 nm with a microplate reader (Thermo Labsystem Multiskan MK3, Thermo, Waltham, MA, USA), and hemolysis of D2 DDCs was calculated according to the following formula:(1)$$\mathrm{Hemolysis}\;\left(\%\right)=\frac{\mathrm{abs}\;\mathrm{of}\;\mathrm{sample}-\mathrm{abs}\;\mathrm{of}\;\mathrm{negative}\;\mathrm{control}}{\mathrm{abs}\;\mathrm{of}\;\mathrm{positive}\;\mathrm{control}-\mathrm{abs}\;\mathrm{of}\;\mathrm{negative}\;\mathrm{control}}\times 100\%$$

### 2.6. In Vitro Release Studies of Two Model Drugs

In vitro release of Bim and DOX from R1 or R2 DDSs was determined in dialysis bags after incubation in PBS for 8 weeks. In brief, R1 or R2 DDSs weighing 29 mg (i.e., the weights of Bim and DOX were 1.0 mg and 0.5 mg according to Table 2) were sealed into a regenerated cellulose dialysis bag (MD25-3500, Viskase, Lombard, IL, USA) and then immersed in 12.5 mL of PBS with shaking (150 rpm) at 37 °C. Meanwhile, 27.5 mg of D2 DDCs without loading drugs was taken as a control. At predetermined time points (2, 4, 8, 12, and 24 h, and 2, 3, 4, 5, 7, 10, 14, 18, 23, and 28 days), for each sample, 2 mL of the supernatant was collected and replaced with an equal amount of fresh PBS. The concentrations of Bim and DOX were determined using a UV/Vis/NIR spectrophotometer (GENESYS 180, Thermo, Waltham, MA, USA) at 270 nm and 480 nm. In the present study, the drug weights after 8 week release were regarded as the actual drug content, which was used for evaluating the encapsulation efficiency of the two model drugs.

### 2.7. Cell Experiments

#### 2.7.1. Cell Culture

Human umbilical vein endothelial cells (HUVECs) were isolated from human umbilical cord veins collected from Peking University People’s Hospital (Beijing, China). Informed consent was provided according to the Declaration of Helsinki. The published method was used to extract and identify HUVECs from umbilical cord vein tissue [39]. A549 cells were obtained from Institute of Basic Medical Sciences (IBMS) of Chinese Academy of Medical Sciences (CAMS). HUVECs or A549 cells were cultured in RPMI-1640 culture medium (Beijing Solarbio Science & Technology Co., LTD, Beijing, China) supplemented with penicillin (100 U/mL), streptomycin (100 g/mL), and 10% *v*/*v* fetal bovine serum (FBS, Zhejiang Tianhang Biotechnology Co., LTD., Huzhou, China), in a humidified incubator in an atmosphere of 37 °C and 5% CO_2_.

#### 2.7.2. Cytotoxicity

HUVECs (4 × 10^3^ per well) or A549 cells (8 × 10^3^ per well) were seeded in 96-well culture plates. After 24 h, 100 μL of culture medium containing D2 DDCs with concentrations of 3.125, 6.25, 12.5, 25, 50, 100, 200, 400, or 800 μg/mL was added to each well for another 24 h culture. The in vitro cell viability was measured with a Cell Counting Kit-8 (CCK-8, Dojindo Laboratories, Kumamoto, Japan) in accordance with the manufacturer’s instructions. Briefly, after washing with PBS twice, 10 μL of CCK-8 reagent solution and 90 μL of cell culture medium were added to each well. The absorbance was measured at 450 nm with a microplate reader after 2 h incubation.

#### 2.7.3. Magnetic Responsiveness Assay

For the magnetic responsiveness assay, 200 μg/mL of D2 DDCs were incubated with A549 cells at 37 °C for 24 h. Subsequently, a ferrite magnet (98 × 48 × 5 mm, 0.5 T) was placed near one edge of the Petri dish for 2 h, and then cells were imaged using a fluorescent microscope.

#### 2.7.4. Cell Imaging Capability Assay

After A549 cells were seeded in 6 cm Petri dishes and grew for 24 h as a monolayer, 4 mL of cell culture medium containing D2 DDCs (200 μg/mL) was added. Images were taken using a fluorescent microscope after incubating for another 24 h culture to evaluate the cell imaging capability of D2 DDCs.

#### 2.7.5. Drug Release for Killing A549 Cells

A549 cells were seeded in a 96-well plate at a density of 4 × 10^3^ cells per well and cultured for 24 h. R2 DDSs were added to the cell culture medium to achieve different concentrations of 3.125, 6.25, 12.5, 25, 50, 100, 200, 400, and 800 μg/mL. A CCK-8 assay was performed after 48 h coculture of R2 DDSs and A549 cells.

Furthermore, A549 cells (2 × 10^4^ cells per well) were seeded in 24-well culture plates, followed by adding 500 μL of culture medium and co-culturing with D2 DDCs (100 μg; initial concentration was 200 μg/mL), R1 DDSs (100 μg), R2 DDSs (100 μg), or drugs (2.76 μg of Bim and 1.21 μg of DOX; the same weight contained in 100 μg of R1 or R2 DDSs) for 1, 2, 4, 7, 10, 14, 21, and 28 days. During the process, 300 μL of culture medium was added to each well every 3 days, and cell viabilities were tested by CCK-8.

### 2.8. Characterizations

All of the quantitative data were reported as means ± standard deviations (SD). The statistical analysis was performed by using the software SPSS 19.0 at a confidence level of 95%. The groups were compared by one-way analysis of variance (ANOVA). A probability value (*p*) of less than 0.05 was considered to be statistically significant.

## 3. Results

### 3.1. Preparation of NPs

As shown in the TEM image in Figure 2A, Fe_3_O_4_ MNPs presented a uniform spherical morphology. With an average value of 136 ± 38 nm, the diameter of Fe_3_O_4_ MNPs was mainly distributed between 100 and 200 nm. The zeta potential of Fe_3_O_4_ MNPs was +19.5 mV (Figure 2F), indicating the good dispersibility and stability of positively charged MNPs in water. As shown in Figure 2B, Fe_3_O_4_ MNPs uniformly dispersed in PLGA/THF solution could be quickly attracted to the side where a magnet was placed. The saturation magnetic induction intensity of Fe_3_O_4_ MNPs was 77.2 emu/g, which reflected the excellent magnetic property of Fe_3_O_4_ MNPs.

NaYF_4_:Eu^3+^ NPs prepared using a hydrothermal method were roughly spherical with a diameter of 32 ± 6 nm (Figure 2C). As shown in Figure 2F, the zeta potential of NaYF_4_:Eu^3+^ NPs was +30.3 mV, obviously higher than that of Fe_3_O_4_ MNPs, suggesting that NaYF_4_:Eu^3+^ NPs had a better stability in water. This might be due to the fact that Fe_3_O_4_ MNPs were not only much larger than NaYF_4_:Eu^3+^ NPs in particle size, but also possessed magnetic properties, which could easily cause self-aggregation. Subsequently, the luminescent property of NaYF_4_:Eu^3+^ NPs was studied under ultraviolet light, and strong red luminescence was observed (Figure 2D). Furthermore, the emission spectrum of NaYF_4_:Eu^3+^ NPs at the excitation wavelength of 394 nm [40] was evaluated. Two obvious emission bands were centered at 593 and 615 nm, corresponding to ^5^D_0_ → ^7^F_1_ and ^5^D_0_ → ^7^F_2_ transitions of Eu^3+^ ions, respectively.

As shown in Figure 2E, CS NPs prepared by electrospraying showed a spherical morphology, and the diameter of CS NPs was 145 ± 45 nm. Due to the repulsion of cations in the chitosan chains [41] and poor volatility of the solvents (the main component was water), it was difficult to obtain CS NPs via electrospraying. To successfully prepare CS NPs, a high concentration of acetic acid solution (70%) was used to reduce the repelling effect of cations and increase the volatilization, and a high voltage of 20 kV was set to generate a strong electric field force. The zeta potential of CS NPs was +11.3 mV (Figure 2F), suggesting that CS NPs were positively charged. The larger CS NPs had a lower zeta potential, indicating the relatively poorer stability compared to Fe_3_O_4_ MNPs and NaYF_4_:Eu^3+^ NPs.

### 3.2. Morphological Observation

As shown in Figure 3A, S1, D1, S2, and D2 DDCs were all rod-shaped in morphology and uniform in size. Moreover, S1 and S2 DDCs prepared by single-needle electrospraying were thinner than D1 and D2 DDCs prepared by double-needle electrospraying. In addition, it was found that the surfaces of S2 and D2 DDCs were rougher than those of S1 and D1 DDCs, which might have been caused by the increase in the number of embedded NPs.

The lengths and diameters of at least 400 of DDCs in each group were measured to further explore the size difference of different DDCs. As shown in Appendix A, the length and diameter of each DDC were basically normally distributed. Specifically, the average lengths of S1, D1, S2, and D2 DDCs were 12.6 ± 2.9, 17.4 ± 4.2, 13.6 ± 3.4, and 19.6 ± 4.4 μm, and the average diameters were 1.2 ± 0.3, 1.7 ± 0.3, 1.3 ± 0.3, and 1.9 ± 0.3 μm, respectively (Figure 3B). It could be observed that the length and diameter of S1 or D1 DDCs were smaller than those of S2 or D2 DDCs embedded with CS NPs, respectively, with the chamber structure obviously showing a great impact on the size of DDCs. The sizes of D1 or D2 DDCs with Janus double chambers were approximately 40% and 44% larger than those with a single chamber (S1 or S2 DDCs), respectively.

Furthermore, the aspect ratios (the ratio of length to diameter) of S1, D1, S2, and D2 DDCs were also calculated, which were interestingly all around 10.5 (Figure 3C). According to our previous study [18], the aspect ratios of rod-shaped particles follow the surface wave theory, i.e., 4.51⋅*N* (*N* = 1, 2, 3, …). In this study, under the same electrospray parameters (especially the precursor solution concentration, PLGA molecular weight, and solvent types), the aspect ratios of rod-shaped DDCs were calculated with *N* = 2 or 3, which were influenced neither by the electrospray method (single needle or double needles) nor by the loading of CS NPs. Zhang et al. [22] prepared 2–10 μm long DOX-loaded short fibers for cancer therapy by intravenous injection; thus, the size of DDCs in this study might be slightly large for cancer treatment. Directly injecting DDSs into the lesion site might facilitate the use of large-size DDCs for cancer treatment, which could also arouse interests in the applications of DDCs for treating diseases such as arthritis and repairing tissue defects by loading growth factors. In fact, using different solvents or adding salts could decrease the size of rod-like DDCs prepared by electrospraying [18]. Therefore, the method of preparing rapeseed pod-like carriers via double-needle electrospraying could not only provide an important reference for fabricating multifunctional DDCs, but also be used to fabricate carriers in different sizes to adapt to more applications, such as intravenous injection, in situ injection, and composites with other materials.

### 3.3. Physicochemical Characterizations

The FTIR spectra of PLGA, CS, and D2 DDCs are presented in Figure 4A. Three strong absorption peaks at 1750, 1167, and 1085 cm^−1^ represented the characteristic carbonyl peaks and the asymmetric and symmetrical stretching vibrations of C–O–C belonging to PLGA. The broad peak near 3414 cm^−1^ and the weak peak at 1616 cm^−1^ were assigned to the stretching vibration of O–H and the in-plane shear vibration of N–H from CS, respectively. Apparently, the characteristic peaks of PLGA and CS all appeared in the FTIR spectrum of D2 DDCs. Among them, some characteristic peaks of CS were relatively weak due to the low content. No new absorption peak was observed in the infrared spectrum of D2 DDCs, indicating that embedding CS NPs in PLGA during electrospraying was only a physical process. The contact angles of D1 DDCs and D2 DDCs were 84.1° ± 0.6° and 83.3° ± 0.8°, respectively, lower than 90°, indicating that the magnetic–luminescent bifunctional rapeseed pod-like DDCs showed a certain degree of hydrophilicity. The contact angles of D1 DDCs and D2 DDCs did not show a significant difference, with both being obviously higher than the contact angle of CS NPs (49.4° ± 2.2°), but not influenced by the embedding of CS NPs.

The components of Fe_3_O_4_ MNPs and NaYF_4_:Eu^3+^ NPs in D2 DDCs were further analyzed by X-ray diffraction. As shown in Figure 4C, the diffraction peaks at 30.1°, 35.5°, 43.1°, 57.0°, and 62.7° were indexed to the cubic phase of Fe_3_O_4_ (JCPDS card No. 74-0748). The strong absorption peaks at 24.3°, 25.8°, 27.6°, 30.9°, 43.7°, 46.7°, 47.5°, and 48.9° corresponded to the standard spectra of cubic and hexagonal phases of NaYF_4_:Eu^3+^ (JCPDS No. 77-2042 and JCPDS No. 16-0334). Moreover, the broad peaks between 10° and 20° belonged to the amorphous material PLGA. These results suggest that Fe_3_O_4_ MNPs and NaYF_4_:Eu^3+^ NPs were successfully embedded in rapeseed pod-like DDCs.

As shown in Figure 4D, the Janus dual-chamber structure could be clearly observed under a metallurgical microscope, and the brown area indicates where Fe_3_O_4_ MNPs were present. With an applied field of 10 kOe, the hysteresis loops of S2 and D2 DDCs (Figure 4E) were similar to that of Fe_3_O_4_ MNPs. However, the saturation magnetic induction intensities of S2 and D2 DDSs were only 4.1 and 3.9 emu/g, respectively, due to the low content of Fe_3_O_4_ MNPs in the entire DDSs. Nevertheless, the saturation magnetic induction intensities of the DDCs were still relatively high, indicating that the prepared rapeseed pod-like DDCs possessed good magnetic properties. No significant difference was observed between S2 and D2 DDCs in terms of saturation magnetization induction intensity, indicating that the chamber structure had little effect on the magnetic properties of DDCs.

However, the Janus dual-chamber structure markedly enhanced the luminescence intensity of D2 DDCs, which was approximately three times stronger than that of S2 DDCs (Figure 4F). Obviously, the dual-chamber structure elaborately separated Fe_3_O_4_ MNPs and NaYF_4_:Eu^3+^ NPs and effectively reduced the quenching effect of MNPs on luminescent material. Furthermore, the bright field and fluorescence images of S2 and D2 DDCs are also presented in Figure 4G. For S2 DDCs, the luminescence was almost invisible, while the dual-chamber structure (yellow arrows in Figure 4G, corresponding to Appendix A) of D2 DDCs could be easily observed. It was noticed that not all D2 DDCs showed a dual-chamber structure (blue and green arrows in Figure 4G), which might be due to the random dropping of D2 DDCs on the collector with Fe_3_O_4_ MNPs (Appendix A) or NaYF_4_:Eu^3+^ NPs and CS NPs (Appendix A) facing upward. In summary, Fe_3_O_4_ MNPs and NaYF_4_:Eu^3+^ NPs dispersed in two chambers effectively enhanced the luminescence intensity of the carriers, endowing them with excellent luminescent properties, which was helpful for fluorescent tracing of the DDSs during the process of drug delivery.

### 3.4. Biocompatible Evaluation

Hemocompatibility is one of the important indicators for evaluating biocompatibility [42]. The hemolysis rate of D2 DDCs was measured by using fetal bovine-derived whole blood. As shown in Figure 5A, after coincubation with D2 DDCs for 30 min and centrifugation, the red blood cells were aggregated at the bottom, and no obvious hemoglobin entered the upper clear solution. The hemolysis rates of D2 DDCs at concentrations of 3.125–800 μg/mL were lower than 5% (according to the International Organization for Standardization ISO/TR7405: hemolysis safety means a hemolysis rate below 5% [43]). It suggested that the prepared rapeseed pod-like DDCs had excellent hemocompatibility.

As shown in Figure 5B, the results of cell viability indicated that D2 DDCs had no obvious cytotoxicity to both HUVECs and A549 cells. Specifically, at the concentration of 200 μg/mL or below, the cell viabilities of HUVECs were above 95%. Although a decreasing trend was observed, the cell viabilities were still higher than 90% at a concentration above 200 μg/mL. Commonly, the cell viability of cancer cells is more active compared to normal cells, and A549 cell viabilities with D2 DDCs were all above 95% at the concentration ranging from 3.125 to 800 μg/mL. Because of the excellent biocompatibility of PLGA, it was not surprising for the prepared DDCs to show good cell compatibility.

Furthermore, the magnetic responsiveness of D2 DDCs was also evaluated during the process of cell culture. Originally, D2 DDCs were well distributed in the culture medium (Figure 5Ci). After 24 h of culture, a square magnet was placed on one side of the petri dish for another 2 h. It was found that, on the side far from the magnet, D2 DDCs were relatively sparse (Figure 5Cii), while D2 DDCs were very dense on the side near the magnet (Figure 5Ciii). D2 DDCs still showed excellent magnetic properties after cocultivation with A549 cells, and they can, thus, be applied in magnetic targeted therapy. The cell imaging capability of D2 DDCs was examined using a fluorescence microscope after culture with A549 cells for 24 h (Figure 5Di–iii). Under the excitation of ultraviolet light, it can be seen in Figure 5Dii that D2 DDCs emitted red light. The results suggested that D2 DDCs had superior fluorescence stability, even though they were sterilized by ultraviolet light for 0.5 h before coculture with A549 cells, indicating the potential of rapeseed pod-like DDSs in bioimaging.

### 3.5. Drug Release Evaluation

Taking Bim and DOX as model drugs, DOX was first loaded in CS NPs to produce DOX@CS NPs, which were then used to prepare R2 DDSs ([NaYF_4_:Eu^3+^/(DOX@CS)@PLGA]//[Bim/Fe_3_O_4_@PLGA]) to achieve the sequential release of Bim and DOX. The FTIR spectra in Appendix A indicate that DOX was encapsulated in CS NPs, whereas Bim and DOX were indeed embedded in R2 DDSs. Subsequently, an 8 week in vitro drug release was performed to evaluate the Bim and DOX encapsulation efficiencies of R1 and R2 DDSs. As shown in Figure 6A, the encapsulation efficiencies of Bim in R1 and R2 DDSs were 80.7% ± 3.4% and 81.0% ± 2.6%, respectively, while those of DOX were 71.7% ± 2.4% and 68.3% ± 7.0%, respectively.

For R1 DDSs ([NaYF_4_:Eu^3+^/DOX/CS@PLGA]//[Bim/Fe_3_O_4_@PLGA]), the release of Bim and DOX exhibited similar trends to those shown in Figure 6B,C. After a burst release of approximately 90% Bim and 70% DOX within the initial 12 h, the release of the two drugs entered a slow and plateau phase. When using PLGA to prepare DDCs, the pores between polymer chains were sufficient to allow H_2_O and the small-molecule drugs Bim and DOX to pass. After the DDSs were dispersed in PBS solution, H_2_O could easily penetrate into the DDSs. Owing to the high solubility, the two drugs rapidly diffused from DDSs into the PBS solution, causing the burst release. Although it is suggested that rapid release of a drug into the interstitium from DDSs may cause premature release and systemic side-effects [29], the burst release of drugs might play an important role in treatments that require a large amount of drugs in the initial process, such as wound therapy and targeted therapy.

The release of Bim in R2 DDSs was similar to that in R1 DDSs, showing an initial burst release within 12 h, but the release of DOX changed significantly compared to R1 DDSs (Figure 6D). In the initial 12 h, the DOX in R2 DDSs showed a burst release to approximately 45%, and almost no DOX was released between 12 and 48 h (Figure 6E). DOX began to release slowly again on the third day, and the release continued until the 14th day. Specifically, the release rate of DOX was relatively slow between days 3 and 5, and it became rapid between days 7 and 14. Subsequently, the release of DOX slowed down and reached the maximum value of accumulation release. The initial burst release of DOX might have been caused by some DOX released from DOX@CS NPs into the PLGA/THF precursor solution during the 4 h vigorous stirring process, which was then embedded into the chamber of R2 DDSs. After 3 days, DOX loaded in DOX@CS NPs was gradually released and further diffused from R2 DDSs into the PBS solution.

Obviously, Bim exhibited rapid release rates in R1 and R2 DDSs. The release differences of DOX at every two detection time points were calculated to further evaluate the release rate of DOX, and the results are shown in Figure 6F. Only one sharp peak in the Gauss fitted curve of R1 DDSs suggested that DOX was also released rapidly, similarly to Bim in R1 DDSs, while two significant peaks were shown in the curve of R2 DDSs. The initial high sharp peak corresponded to the initial burst release of DOX, and the broad peak between days 3 and 18 reflected the sustained release of DOX in R2 DDSs. In sum, the prepared rapeseed pod-like R2 DDSs achieved the sequential release of dual drugs.

In vitro cell experiments of magnetic–luminescent bifunctional rapeseed pod-like DDSs were performed to further evaluate the drug release kinetics. Firstly, A549 cells were used to culture with R2 DDSs at different concentrations to determine an appropriate drug concentration. As shown in Figure 7A, after 48 h of coculture, R2 DDSs showed significant cytotoxicity to A549 cells at concentrations above 100 μg/mL, and the cell viability of A549 cells was 23% ± 5% at 800 μg/mL. Since A549 cells were cocultured with R2 DDS for 48 h, during which only part of DOX was released, the concentration of 200 μg/mL with a moderate cell activity was selected for further study.

Subsequently, a 28 day sustained culture of A549 cells in the culture medium dispersed with D2, R1, R2, or drugs (Bim/DOX) was also performed. As shown in Figure 7B, the cell viability of A549 cells cultured with D2 DDCs which contained no drug always remained near 100% within 28 days. On the first day, the cell viabilities in the R1 group (drugs were directly embedded in double chambers) and pure drug (Bim/DOX) group were 53% ± 4% and 24% ± 3%, respectively, which were reduced to approximately 0% on the fourth day. The results were consisted with the initial burst release of DOX in R1 DDSs, also indicating the sufficient dosage of the drugs. For the R2 group (DOX was embedded in DOX@CS NPs), A549 cell viability decreased within the first 2 days, which was attributed to the release of DOX in the first phase. Interestingly, the cell viability rebounded on the fourth day, likely owing to the addition of a fresh culture medium and the end of the first phase of DOX release. After 7 days, with the subsequent continuous release of DOX, A549 cell viability gradually decreased over time and declined to only 23% ± 6% on the 28th day. These results support the sustained release of DOX in the second stage, corresponding to the results in Figure 6D. They suggest that R2 DDSs had excellent performance in maintaining the sustained release of drugs, indicating their potential application in the sustained treatment of diseases.

Mechanisms of dual drugs released sequentially in R2 DDSs are illustrated in Figure 7C according to the results of in vitro drug release evaluation and coculture of A549 cells and DDSs. Within 12 h, Bim and partial DOX, which entered the chamber from DOX@CS NPs during the electrospraying process, were released quickly (Figure 7Cii). On days 3 to 18, the remaining DOX loaded in DOX@CS NPs was first released from DOX@CS NPs into the chamber, and then released from R2 DDSs (Figure 7Ciii). In summary, by embedding DOX-loaded NPs into one chamber of the rod-shaped DDSs and loading with Bim in another chamber (Figure 7Ci), the two drugs could be released in succession, and a sustained release of DOX was also achieved. The combination of a dual-chamber structure and a core–shell structure to achieve the sequential release of dual drugs provides an important reference for the delivery of dual or multiple drugs. The designed rapeseed pod-like DDSs can also be applied to combination medication for cancer treatment and recurrence prevention, as well as the inhibition of inflammation in arthritis and other diseases.

## 4. Conclusions

In this work, a new bionic idea inspired by the dual-chamber structure rapeseed pods and the core–shell structure of rapeseeds containing oil wrapped in rapeseed pods was proposed, allowing the successful establishment of magnetic–luminescent bifunctional rapeseed pod-like DDCs for sequential release of dual drugs via double-needle electrospraying in one step. Similarly to natural rapeseed pods, the prepared [NaYF_4_:Eu^3+^/chitosan (CS)@PLGA]//[Fe_3_O_4_@PLGA] DDCs had good rod-shaped morphology and an obvious Janus dual-chamber structure. In addition to superior magnetic properties, the luminescent properties of DDCs were also significantly enhanced by the Janus structure separating magnetic and luminescent materials. Synchronously, the rapeseed pod-like DDCs also performed splendidly in terms of hemocompatibility and cytocompatibility. By embedding the DOX-loaded NPs into one chamber of rapeseed pod-like [NaYF_4_:Eu^3+^/(DOX@CS)@PLGA]//[Bim/Fe_3_O_4_@PLGA] DDSs and loading Bim into another chamber, the sequential release of the two model drugs was realized. The subsequent sustained release of DOX led to long-lasting and effective killing of cancer cells in vitro at a low drug dosage. The obtained results demonstrated a distinguished effect with the magnetic–luminescent bifunctional rapeseed pod-like DDSs for sequential release of dual drugs. In sum, the DDSs with excellent magnetic and luminescent properties are highly promising for application in clinical drug targeted delivery, bioimaging, and sustained treatment of diseases.

## Figures and Tables

**Figure 1 pharmaceutics-13-01116-f001:**
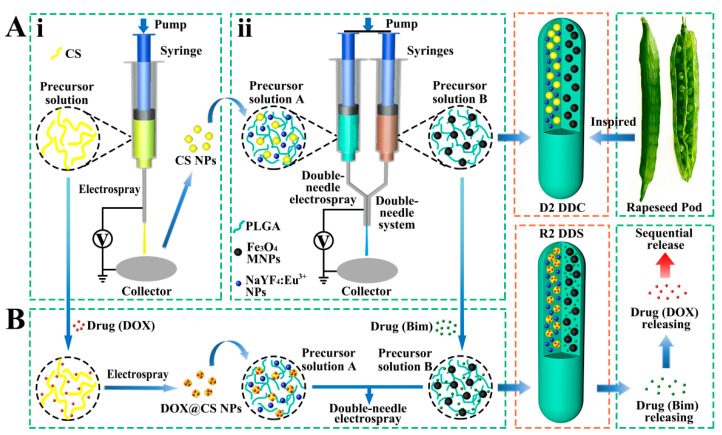
(**A**) Schematic diagram of the process of preparing magnetic–luminescent bifunctional rapeseed pod-like D2 DDCs: (**i**) schematic diagram of the preparation of CS NPs via electrospraying; (**ii**) schematic diagram of preparing D2 DDCs by double-needle electrospraying. (**B**) Schematic diagram of the process of preparing magnetic–luminescent bifunctional rapeseed pod-like R2 DDSs for the sequential release of dual drugs.

**Figure 2 pharmaceutics-13-01116-f002:**
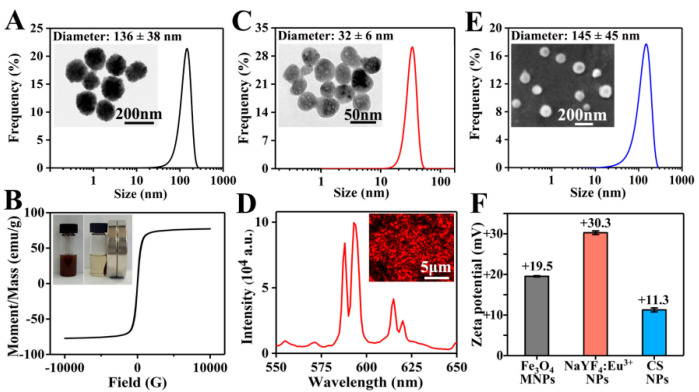
(**A**) TEM image and diameter distribution of Fe_3_O_4_ MNPs; more than 450 Fe_3_O_4_ MNPs were measured. (**B**) Hysteresis loop and magnetic responsiveness testing photographs of Fe_3_O_4_ MNPs. (**C**) TEM image and diameter distribution of NaYF_4_:Eu^3+^ NPs; more than 350 NaYF_4_:Eu^3+^ NPs were measured. (**D**) Fluorescent microscope image and emission spectra of NaYF_4_:Eu^3+^ NPs. (**E**) SEM image and diameter distribution of CS NPs; more than 400 CS NPs were measured. (**F**) Zeta potentials of Fe_3_O_4_ MNPs, NaYF_4_:Eu^3+^ NPs, and CS NPs.

**Figure 3 pharmaceutics-13-01116-f003:**
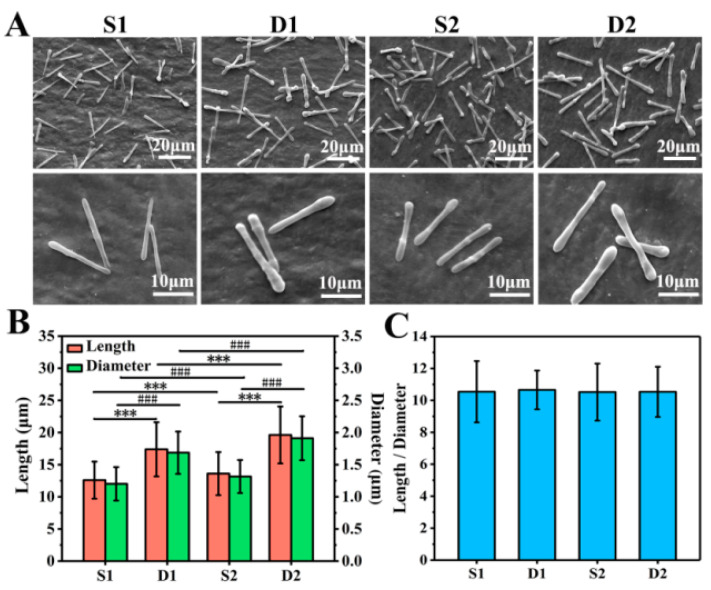
(**A**) SEM images of S1, D1, S2, and D2 DDCs. (**B**) Average lengths and diameters of S1, D1, S2, and D2 DDCs, showing a significant difference in length (*** *p* < 0.001) and in diameter (^###^
*p* < 0.001). (**C**) Ratios of length to diameter of S1, D1, S2, and D2 DDCs; at least 400 of DDCs were observed to measure the length, diameter, and aspect ratio, using the software Image J 1.48v.

**Figure 4 pharmaceutics-13-01116-f004:**
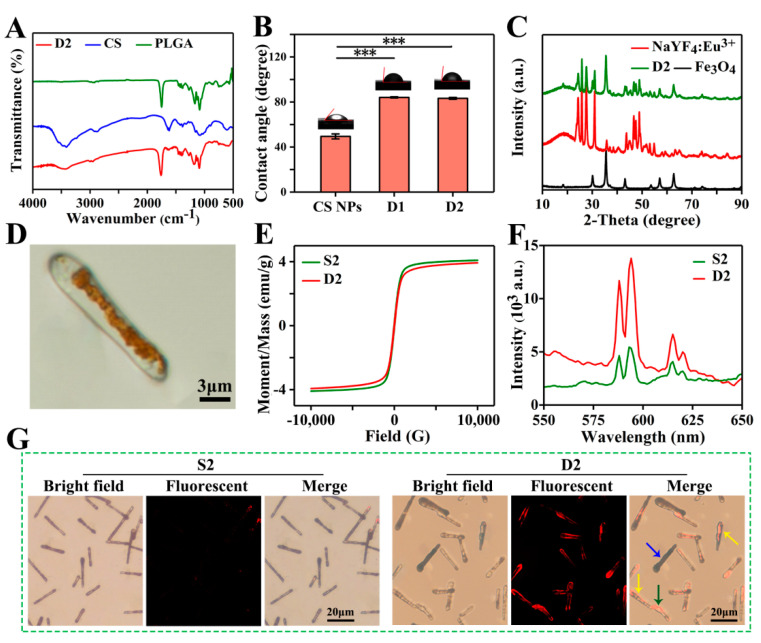
(**A**) FTIR spectra of CS, PLGA, and D2 DDCs. (**B**) Contact angles of CS NPs and D1 and D2 DDCs, and the corresponding images of water droplets on different samples, *** *p* < 0.001. (**C**) XRD patterns of Fe_3_O_4_ MNPs, NaYF_4_:Eu^3+^ NPs, and D2 DDCs. (**D**) Optical microscope image of D2 DDC. (**E**) Hysteresis loops of S2 and D2 DDCs. (**F**) Emission spectra of S2 and D2 DDCs. (**G**) Fluorescent microscope images of S2 and D2 DDCs.

**Figure 5 pharmaceutics-13-01116-f005:**
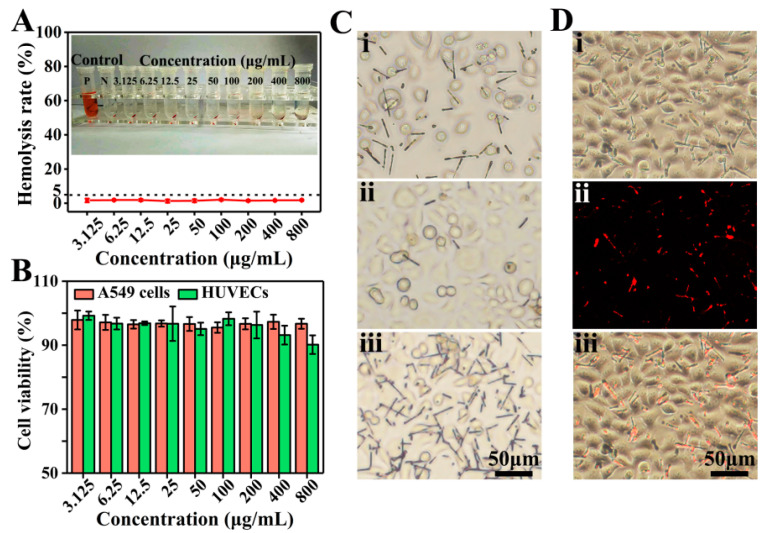
(**A**) Hemolysis rate and photographs of D2 DDCs at concentrations ranging from 3.125 to 800 μg/mL. (**B**) In vitro cell viabilities of A549 cells and HUVECs normalized to the untreated control after incubation with D2 DDCs for 24 h at concentrations of 3.125–800 μg/mL. (**C**) Optical microscope images of A549 cells incubated with the D2 DDCs under the effect of a magnet: (**i**) without magnet; (**ii**) far from the magnet; (**iii**) near the magnet. (**D**) Fluorescent microscope images of A549 cells incubated with D2 DDCs: (**i**) bright-field image; (**ii**) fluorescent image; (**iii**) overlay of the bright-field and fluorescent images.

**Figure 6 pharmaceutics-13-01116-f006:**
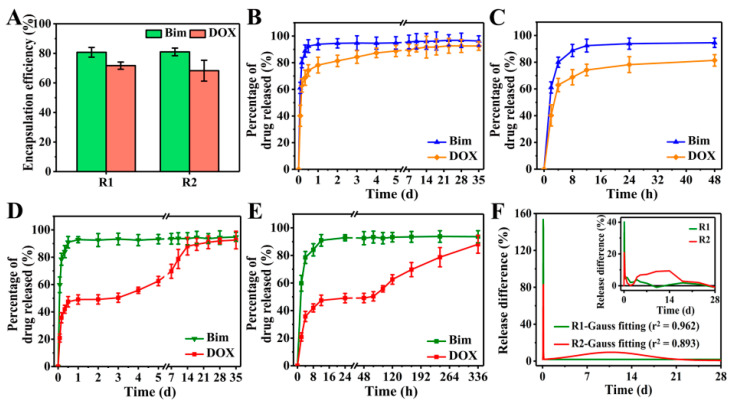
(**A**) Bim and DOX encapsulation efficiencies of R1 and R2 DDSs. Percentage of Bim and DOX released from R1 DDSs within (**B**) 35 days and (**C**) 48 h. Percentage of Bim and DOX released from R2 DDSs within (**D**) 35 days and (**E**) 336 h. (**F**) Release difference of DOX in R1 and R2 DDSs measured at every two detection time points: measured value (inset) and Gauss fitted value.

**Figure 7 pharmaceutics-13-01116-f007:**
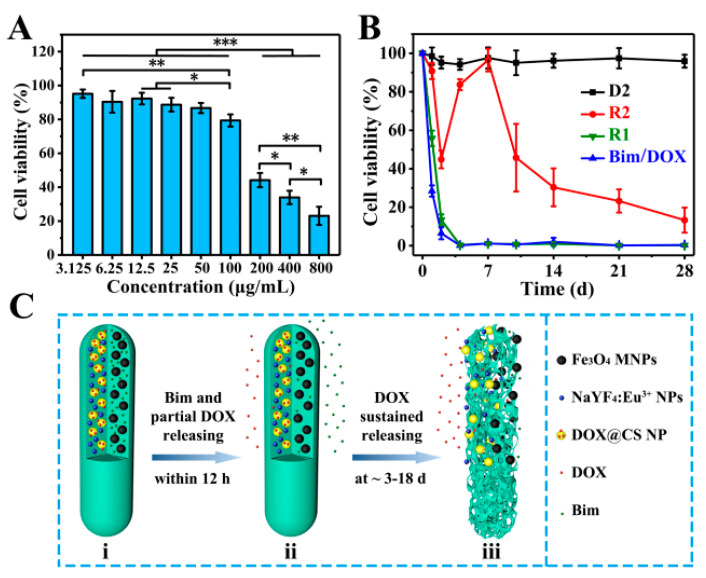
(**A**) In vitro A549 cell relative viabilities normalized to the untreated control after incubation for 48 h with different concentrations of R2 DDSs; * *p* < 0.05, ** *p* < 0.01, *** *p* < 0.001. (**B**) In vitro A549 cell relative viabilities normalized to the untreated control for 28 days with D2 DDCs, R1 and R2 DDSs, and drugs of Bim and DOX. (**C**) Schematic illustration of the release of two model drugs (Bim and DOX) from R2 DDSs: (**i**) R2 with Bim in left chamber and DOX@CS NPs in right chamber before drug release; (**ii**) Bim and partial DOX released within 12 h; (**iii**) the suspended release of DOX within ~3 to 18 days.

**Table 1 pharmaceutics-13-01116-t001:** Compositions of the precursor solutions for preparing different DDCs.

DDC	Precursor Solution	Fe_3_O_4_ MNPs (mg)	NaYF_4_:Eu^3+^ NPs (mg)	CS NPs (mg)	PLGA (mg)	THF (mL)
S1	-	50	50	-	150	5.0
D1	A	-	50	-	75	2.5
	B	50	-	-	75	2.5
S2	-	50	50	25	150	5.0
D2	A	-	50	25	75	2.5
	B	50	-	-	75	2.5

**Table 2 pharmaceutics-13-01116-t002:** Compositions of the precursor solutions for preparing R1 and R2 DDSs.

DDC	Precursor Solution	Fe_3_O_4_ MNPs (mg)	NaYF_4_:Eu^3+^ NPs (mg)	Drug (Bim)	Drug (DOX)	PLGA (mg)	THF (mL)
R1	A	-	50	-	DOX: 5mg, CS NPs: 25mg	75	2.5
	B	50	-	10 mg	-	75	2.5
R2	A	-	50	-	DOX@CS NPs: 30 mg	75	2.5
	B	50	-	10 mg	-	75	2.5

## Data Availability

The data presented in this study are available within the article and its supplementary materials.

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
