# Peer review of "Preparation of Magnetic–Luminescent Bifunctional Rapeseed Pod-Like Drug Delivery System for Sequential Release of Dual Drugs†"

_pharmaceutics, 2021, doi:10.3390/pharmaceutics13081116_

Round 1

Reviewer 1 Report

  1. Abstract; The results of the study should be highlighted. Additionally, some quantity results must be added to this section.
  2. Introduction; Line 56-58; Depending on the contact of the nanorod with the cell surface, the internalization process may be difficult. It is recommended to read the following article for more information: https://doi.org/10.1016/j.nantod.2020.101057
  3. Introduction; Line 71-74; Slow-release (Sustained release) can increase the effects of multi-drug resistance. I suggest you read the following article: https://doi.org/10.1016/j.jconrel.2020.08.012
  4. What is the main contribution of this manuscript compared to previously published ones?
  5. Add more discussion about the results extracted from figures. This is one of the main shortcomings of this manuscript.
  6. Results-Morphological observation; Line 312-315; The size of the carrier is large, so limited its applications, especially in anti-cancer applications. Discuss in this regard.
  7. Results-Drug release evaluation:
  • There is no logical explanation for what happened. 
  • In response to which stimulus is the drug released?
  • If released without response, the carrier is shown to be unstable, which will lead to various side effects.
  • In Figure 6 (B-E), the vertical axis caption is not "drug release rate" but "percentage of drug released". However, it is suggested that additional explanations be added regarding the release rate.

Round 2

Reviewer 1 Report

The comments are applied in an acceptable way. Now, the paper has the worth of publication. 

Author Response

Thank you very much for your careful review and evaluation of this work. 

Reviewer 2 Report

The authors are thanked to have address the comments of the reviewers strengthening their manuscript. I would suggest publication of the revised manuscript after the minor modifications listed below:

  • line 44: ”response” instead of “responce”
  • line 545: please do not start a sentence by “and”
  • line 626: please use only significant numbers: 136 ± 38 nm for example (same for lines 635, 645, 721-723, 776, 780, 917-918, 988, 995-996, 1003
